# 3D-ATRES: Ambiguity-Tolerant Learning for 3D Referring Expression Segmentation

## Abstract

3D Referring Expression Segmentation (3D-RES) is an emerging yet challenging task at the interaction of 3D vision and language, which aims to precisely segment a target instance within a 3D point cloud based on a given natural language referring expression. However, most previous methods overlook multi-source ambiguities that are prevalent in real-world scenarios, including prompt, spatial, and annotation ambiguities. Prompt ambiguity arises from confusion between referent and target instances due to ambiguous language, spatial ambiguity results from viewpoint variations causing incomplete segmentation, and annotation ambiguity stems from inconsistent or noisy labeling in training data. In this paper, we propose a novel 3D Ambiguity-Tolerant Referring Expression Segmentation (3D-ATRES), which explicitly models and mitigates multi-source ambiguities in 3D-RES. Specifically, we employ $TR^2$ Semantic Structurizer to transform free-form natural language into structured Target-Relation-Referent triples, thereby eliminating referential ambiguity. For spatial ambiguity, we introduce a Normal-Aware Spatial Alignment that leverages surface normal cues to achieve viewpoint-consistent geometry alignment. To mitigate annotation ambiguity, we introduce an Annotation Ambiguity Penalty, which enables the network to adaptively learn from noisy or inconsistent annotations through confidence evaluation. Experiments on ScanRefer and Multi3DRefer show that 3D-ATRES achieves state-of-the-art performance, confirming the effectiveness of modeling ambiguity in 3D-RES. The code is available at `https://anonymous.4open.science/r/3D-ATRES`.

## 1 Introduction

The rapid development of 3D vision (Qi et al., 2017b; He et al., 2024) and natural language understanding (Devlin et al., 2019; Radford et al., 2021) has fostered increasing interest in 3D Referring Expression Segmentation (3D-RES), the task that aims to precisely segment a target instance in a point cloud based on a given natural language referring expression. Unlike conventional 3D semantic (Wang et al., 2025b; Qi et al., 2017a; Wang et al., 2025a) or instance segmentation (Takmaz et al., 2023; Peng et al., 2023), 3D-RES requires fine-grained alignment between linguistic descriptions and geometric structures, enabling human-machine interaction in diverse applications such as robotic manipulation (Chen et al., 2023; Kong et al., 2023), AR/VR scene understanding (Wang et al., 2025a; Qiu et al., 2023), and autonomous navigation (Sun et al., 2023).

Despite recent advances (Deng et al., 2025; Zhu et al., 2024b; Wu et al., 2024c), real-world 3D-RES remains highly challenging due to the prevalence of multi-source ambiguities, including prompt, spatial, and annotation ambiguities. First, prompt ambiguity occurs when vague or under-specified language descriptions fail to uniquely identify the target object, causing confusion with visually or semantically similar distractors. As shown in Figure 1(a), this challenge intensifies when the target and referent belong to the same semantic category (e.g., both are chairs). In the scene with multiple chairs, a vague description such as "another chair" is prone to ambiguity, making it difficult to distinguish the intended object. Such cases often mislead less robust models into selecting an incorrect instance. Second, spatial ambiguity arises from viewpoint variations, occlusions, and incomplete scans that compromise geometric understanding. These factors can cause semantically coherent objects to appear as disconnected components in 3D space. As shown in Figure 1(b), a white table's top and legs may be perceived as geometrically separate parts, making it difficult for models to

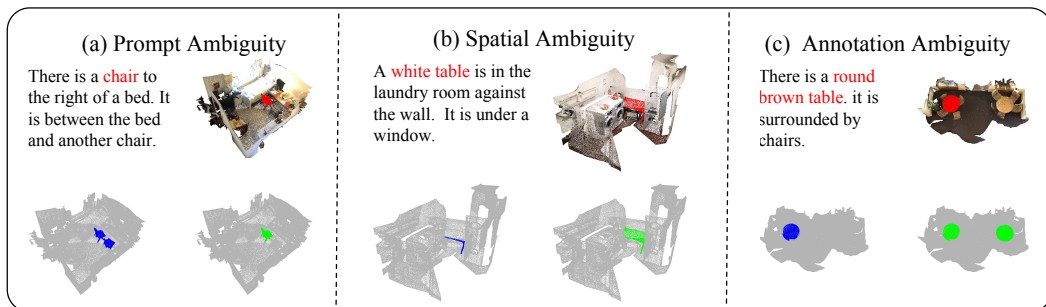

Figure 1: Three pervasive yet underexplored ambiguities in 3D-RES: prompt ambiguity, spatial ambiguity, and annotation ambiguity. Red masks indicate ground truth, green masks represent predictions solely by our method, and blue masks represent predictions solely by the baseline (Deng et al., 2025).

produce a complete segmentation of the entire object. Finally, annotation ambiguity can arise from inevitable errors in manual labeling, especially in large and cluttered scenes. It is common for a referring expression to accurately describe multiple objects, yet the ground truth only identifies one. As shown in Figure 1(c), the instruction could plausibly refer to both tables in the scene, but only a single one is annotated, leading to incomplete and ambiguous supervision.

To overcome these multi-source ambiguities, we propose 3D Ambiguity-Tolerant Referring Expression Segmentation (3D-ATRES), a novel method that explicitly models and mitigates multi-source ambiguities through three key designs: First, we propose the $TR^2$ Semantic Structurizer ($TR^2SS$) to address prompt ambiguity by transforming free-form referring expressions into structured Target-Relation-Referent triples. This explicit decomposition disambiguates the relationships between objects, enabling precise identification of the intended target even in the presence of multiple similar objects. Second, we propose Normal-Aware Spatial Alignment (NASA) to handle spatial ambiguity. By incorporating surface normal information as geometric priors, NASA ensures viewpoint-consistent segmentation and helps the model correctly group disconnected components into semantically complete objects. Third, we introduce the Annotation Ambiguity Penalty (AAP) to address the impact of inconsistent ground-truth labels. AAP leverages the loss distribution of points to estimate annotation reliability, dynamically down-weighting ambiguous or noisy labels during training, allowing the network to prioritize more trustworthy supervision signals.

We evaluate 3D-ATRES on two widely used 3D-RES benchmarks, ScanRefer (Chen et al., 2020) and Multi3DRefer (Zhang et al., 2023), achieving substantial improvements over state-of-the-art methods. Qualitative results further confirm that our method produces precise and robust segmentation even when linguistic descriptions are vague, viewpoints vary significantly, or labels are noisy.

Our contributions can be summarized as follows:

- We reveal and study three sources of ambiguity in 3D-RES: prompt, spatial, and annotation ambiguities, which are prevalent but overlooked in existing work.
- We propose 3D-ATRES, a framework that integrates $TR^2SS$, NASA, and AAP to effectively address multi-source ambiguities in 3D-RES.
- Our method achieves state-of-the-art results on standard benchmarks while maintaining strong robustness in real-world scenes with multi-source ambiguities.

## 2 RELATED WORK

### 2.1 3D VISUAL GROUNDING

3D Visual Grounding (3DVG) aims to locate a target object in a 3D scene based on a natural language description (Chen et al., 2022; Lin et al., 2023; Wu et al., 2023). It can be divided into two categories: Referring Expression Comprehension (REC) and Referring Expression Segmentation

(RES) (Qian et al., 2024b). In REC, the task is to predict a bounding box for the target object referred to by the language expression (Chen et al., 2025a). Existing approaches can be broadly categorized into two-stage (Chen et al., 2022; Feng et al., 2021; He et al., 2021; Yuan et al., 2021; Zhao et al., 2021; Yang et al., 2023; Wu et al., 2024c; Zhang et al., 2023) and single-stage (Luo et al., 2022; Wang et al., 2023a) methods. The two-stage methods first generate object proposals using a detector (Qi et al., 2019) and then select the most relevant one. In contrast, single-stage methods enable end-to-end training (Luo et al., 2022; Jain et al., 2022; Wu et al., 2023; 2024b). Unlike REC, RES is a more challenging task that requires finer-grained visual-linguistic alignment to segment the target object at the mask level (Chen et al., 2025b). Several methods, such as TGNN (Huang et al., 2021) and 3D-STMN (Wu et al., 2024c), have developed two-stage and single-stage frameworks for RES. In addition, a number of notable works (Chen et al., 2025a; Wu et al., 2024a;b; He & Ding, 2024; He et al., 2024; Qian et al., 2024a) have focused on enhancing the fusion of linguistic and visual information. While existing works attempt to address REC and RES tasks, they often overlook the issues of referring, spatial, and annotation ambiguities, which limit their performance in real-world scenarios. Therefore, this paper introduces an ambiguity-aware method enabling more robust 3D RES under real-world conditions.

## 2.2 3D-RES BASED ON MLLMs

With the rapid advancement of Multimodal Large Language Models (MLLMs), a growing body of research has begun to investigate their capabilities in spatial perception and reasoning (Chen et al., 2024a;b; Wang et al., 2023b; Huang et al., 2023; 2024; Hong et al., 2023; Fu et al., 2024; Qi et al., 2025; Zhu et al., 2024b; Zheng et al., 2025). However, existing MLLMs based on 2D images or videos still underperform in 3D tasks such as RES due to their lack of comprehensive 3D spatial understanding (Liu et al., 2024a; Peng et al., 2023). To address this limitation, several studies have integrated 3D information into MLLMs to enhance their spatial cognition. For instance, 3D-LLaVA incorporates more advanced cross-modal modules to improve the fusion of multi-modal features, thereby enriching scene representations (Deng et al., 2025). Similarly, LLaVA-3D utilizes RGB and depth images as representations of 3D structure (Zhu et al., 2024b). Although these works attempt to tackle 3D tasks using MLLMs, they do not explicitly guide the models to resolve inherent ambiguities such as referential, spatial, or annotation uncertainties. In contrast to the aforementioned approaches, our work focuses on leveraging the strong reasoning abilities and generalization of MLLMs to help address these challenges, achieving an optimal balance between efficiency and accuracy in 3D RES tasks.

## 3 METHODOLOGY

Our 3D-ATRES systematically addresses multi-source ambiguities in 3D-RES, as illustrated in Figure 2. In Section 3.1, we first give an introduction to 3D-RES. In Section 3.2, we introduce the $TR^2$ Semantic Structurizer that parses the referring expression into a structured triple to resolve prompt ambiguity. Then, we introduce the Normal-Aware Spatial Alignment in Section 3.3. In Section 3.4, we discuss how the Annotation Ambiguity Penalty models label uncertainty to produce ambiguity-tolerant segmentation masks.

## 3.1 PROBLEM DEFINITION

Given a point cloud scene $\mathcal{S} = \{(\mathbf{x}_i, \mathbf{h}_i)\}_{i=1}^{N}$, where $\mathbf{x}_i$ denotes the spatial coordinates and $\mathbf{h}_i$ represents per-point features (e.g., RGB color, surface normals), and a natural language referring expression $\mathcal{U} = \{\mathbf{w}_1, \mathbf{w}_2, \ldots, \mathbf{w}_L\}$ of length $L$, the goal of 3D-RES is to predict a binary segmentation mask $\mathbf{M}$ that identifies the target object:

$$\mathbf{M} = f_\theta(\mathcal{S}, \mathcal{U}), \tag{1}$$

where $f_\theta(\cdot)$ represents our 3D-ATRES model and $\mathbf{M}_i = 1$ if point $i$ belongs to the referred object, 0 otherwise.

## 3.2 $TR^2$ SEMANTIC STRUCTURIZER

Prior work (Liu et al., 2024b) has shown that hierarchically structuring context sharpens LLM attention, improves key-information retrieval, and enhances complex reasoning. To address prompt am-

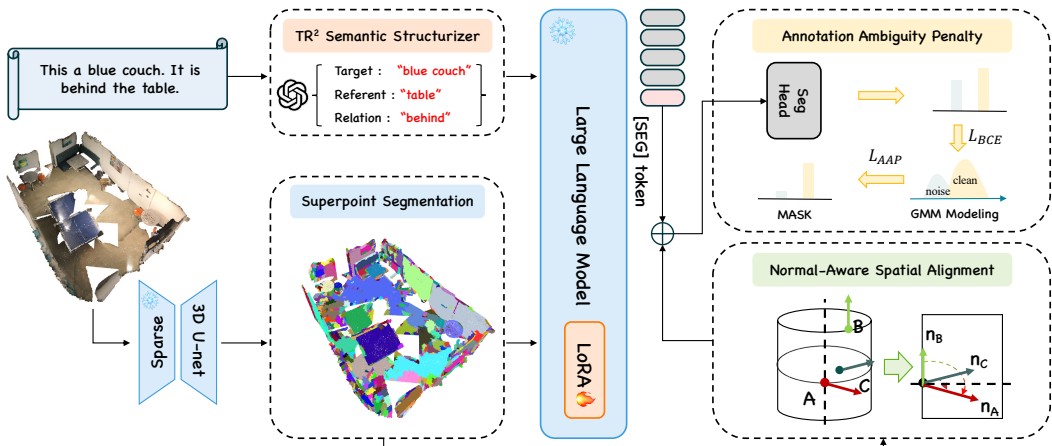

Figure 2: Framework of the proposed 3D-ATRES. First, the TR$^2$ Semantic Structurizer decomposes each referring expression into three token types: Target, Referent, and Relation. A sparse 3D U-net extracts visual features from the point cloud. The visual feature is then clustered into superpoint with Superpoint Pooling. These language token and visual features are fused with LLaVA to produce joint multi-modal representations. Next, a Normal-Aware Spatial Alignment refines these representations, scoring candidate objects according to surface normals and spatial relationships. Finally, a GMM-based Annotation Ambiguity Penalty attenuates the influence of noisy labels.

biguity, we leverage a pre-trained LLM to transform free-form referring expressions into structured semantic representations. Specifically, we employ few-shot prompting to extract Target–Relation–Referent triples:

$$\mathcal{U} \xrightarrow{\text{LLM}} \mathcal{T} = \{(\mathbf{t}_i, \mathbf{r}_i, \mathbf{ref}_i)\}_{i=1}^{K}, \qquad (2)$$

where $\mathbf{t}_i$ denotes the target object, $\mathbf{r}_i$ the spatial relation, $\mathbf{ref}_i$ the reference object, and $K$ is the number of extracted triples. This structured representation explicitly disambiguates targets by encoding their relational context with other objects in the scene.

For non-relational expressions like "the red cabinet", the "relation" and "referent" fields are set to null. Complex expressions (e.g., "the chair in front of the table and beside the window") yield multiple triples capturing different spatial relationships. The extracted triples are then encoded into a unified linguistic representation:

$$\mathbf{F}_{\text{text}} = f_{\text{text}}(\mathcal{T}), \qquad (3)$$

where $f_{\text{text}}(\cdot)$ is a pre-trained text encoder. This structured parsing enables precise target identification even in scenes with multiple similar objects.

### 3.3 NORMAL-AWARE SPATIAL ALIGNMENT

Spatial ambiguity in 3D-RES often arises when object surfaces exhibit large variations in local orientation due to curvature, edges, partial occlusion, or missing regions. Naively, such geometric discontinuities can be misinterpreted as boundaries between different instances, fragmenting the segmentation. To address this, we propose Normal-Aware Spatial Alignment (NASA), which leverages surface normal information to guide geometry-aware feature learning.

Given a point cloud $\mathcal{S} = \{(\mathbf{x}_i, \mathbf{c}_i, \mathbf{n}_i)\}_{i=1}^{N}$ with coordinates $\mathbf{x}_i$, colors $\mathbf{c}_i$, and unit normals $\mathbf{n}_i$, we first extract point features using a sparse 3D U-Net encoder:

$$\mathbf{F}_{\text{geo}} = f_{\text{geo}}(\mathcal{S}). \qquad (4)$$

For any two points $i$ and $j$, we quantify their geometric dissimilarity through the angular deviation between surface normals:

$$\theta_{ij} = \arccos(\mathbf{n}_i^\top \mathbf{n}_j) \in [0, \pi]. \qquad (5)$$

The core insight is that same-instance points with large $\theta_{ij}$ (e.g., table top vs. table leg) should maintain similar features despite their geometric differences. We design an adaptive alignment mechanism that normalizes feature similarities based on geometric context:

$$s_{ij} = \frac{\mathbf{f}_i^\top \mathbf{f}_j}{\|\mathbf{f}_i\|\|\mathbf{f}_j\| \cdot \phi(\theta_{ij})}, \quad \mathbf{f}_i, \mathbf{f}_j \in \mathbf{F}_{\text{geo}}, \tag{6}$$

where $\phi(\cdot)$ is a monotonically decreasing function that compensates for geometric dissimilarity. We use $\phi(\theta) = \exp(-\theta)$.

Let $\mathcal{P}(i)$ denote the set of points belonging to the same instance as anchor point $i$. The NASA loss encourages geometry-normalized alignment:

$$\mathcal{L}_{\text{NASA}} = -\frac{1}{N} \sum_{i=1}^{N} \log \frac{\sum_{j \in \mathcal{P}(i)} \exp(s_{ij}/\tau)}{\sum_{k=1}^{N} \exp(s_{ik}/\tau)}, \tag{7}$$

where $\tau = 0.07$ is the temperature parameter.

NASA explicitly accounts for geometric heterogeneity: when two same-instance points have large normal deviation (high $\theta_{ij}$), the normalization factor $\phi(\theta_{ij})$ reduces the required similarity threshold, preventing the model from treating them as different objects. Conversely, geometrically similar points (low $\theta_{ij}$) maintain standard similarity requirements.

Through this geometry-aware alignment, NASA learns representations that are invariant to local surface variations while preserving instance-level semantic coherence, effectively mitigating spatial ambiguity in 3D-RES.

## 3.4 ANNOTATION AMBIGUITY PENALTY

Annotation ambiguity represents a fundamental yet often neglected challenge in 3D-RES. Unlike 2D images, where object boundaries are usually clear, point clouds introduce several challenges. First, varying viewpoints can lead to differing interpretations of object extents. Second, ambiguous referring expressions, such as "the chair" in a scene with multiple chairs, often result in disagreements among annotators. Finally, the irregular and incomplete nature of point clouds makes defining object boundaries difficult. These factors together cause label noise, where annotators provide inconsistent labels for the same points.

Traditional supervised learning assumes all annotations are equally reliable, which becomes problematic when training with noisy labels. Similar to the McGurk effect (McGurk & MacDonald, 1976), where conflicting sensory inputs distort perception, forcing models to learn from conflicting or incorrect supervision signals often results in poor generalization. To address this, we propose Annotation Ambiguity Penalty (AAP), a probabilistic method that automatically identifies and down-weights unreliable annotations during training.

Given the fused linguistic-geometric features $\mathbf{F} = [\mathbf{F}_{\text{text}}, \mathbf{F}_{\text{geo}}]$, our segmentation head produces point-wise predictions:

$$\hat{\mathbf{Y}} = f_{\text{seg}}(\mathbf{F}) \in [0, 1]^N, \tag{8}$$

where $\hat{y}_i \in \hat{\mathbf{Y}}$ represents the predicted probability that point $i$ belongs to the target object. We compute the binary cross-entropy loss with ground-truth labels $y_i \in \{0, 1\}$ as follows:

$$l_i = -y_i \log(\hat{y}_i) - (1 - y_i) \log(1 - \hat{y}_i). \tag{9}$$

Following (Arpit et al., 2017), we leverage the observation that deep networks learn simple patterns before fitting noise. This manifests in the loss distribution: correctly labeled points yield consistently low losses, while mislabeled points produce persistently high losses. We model this bimodal distribution using a Gaussian Mixture Model (McLachlan & Peel, 2000):

$$p(l_i) = \pi_{\text{clean}} \cdot \mathcal{N}(l_i \mid \mu_{\text{clean}}, \sigma_{\text{clean}}^2) + \pi_{\text{noise}} \cdot \mathcal{N}(l_i \mid \mu_{\text{noise}}, \sigma_{\text{noise}}^2), \tag{10}$$

where the clean component ($\mu_{\text{clean}} < \mu_{\text{noise}}$) captures reliable annotations and the noise component models unreliable ones.

We compute each point's posterior probability of being correctly labeled:

$$w_i = p(\text{clean} \mid l_i) = \frac{\pi_{\text{clean}} \cdot \mathcal{N}(l_i \mid \mu_{\text{clean}}, \sigma_{\text{clean}}^2)}{p(l_i)}. \tag{11}$$

This confidence score $w_i \in [0, 1]$ adaptively identifies annotation reliability without manual intervention.

The AAP loss selectively emphasizes reliable supervision:

$$\mathcal{L}_{\text{AAP}} = \frac{1}{N} \sum_{i=1}^{N} w_i \cdot l_i. \tag{12}$$

The GMM parameters $\Theta = \{\mu_{\text{clean}}, \sigma_{\text{clean}}^2, \mu_{\text{noise}}, \sigma_{\text{noise}}^2, \pi_{\text{clean}}\}$ are updated via Expectation-Maximization every $T$ iterations, allowing dynamic adaptation to the evolving loss distribution. Implementation details are provided in Appendix A.1.

Through this probabilistic approach, AAP transforms annotation ambiguity from a hindrance into an opportunity for robust learning. By automatically identifying and down-weighting unreliable labels, it enables effective training even with imperfect annotations—a crucial capability for 3D-RES where obtaining perfect labels is practically infeasible.

**Overall training objective.** The total loss for training 3D-ATRES is:

$$\mathcal{L}_{\text{total}} = \mathcal{L}_{\text{NASA}} + \mathcal{L}_{\text{AAP}}. \tag{13}$$

## 4 EXPERIMENT

### 4.1 EXPERIMENTAL SETTINGS

**Dataset and Metrics.** All experiments are conducted on the ScanNet-v2 dataset (Dai et al., 2017), which provides the underlying 3D geometry for the two most widely-used 3D referring-expression benchmarks: ScanRefer (Chen et al., 2020) and Multi3DRefer (Zhang et al., 2023). ScanRefer comprises 51 k human-written expressions paired with 800 RGB-D scenes; we adhere to its official train/val/test split. Multi3DRefer extends ScanRefer by adding 13k additional queries that require multi-object reasoning, and we evaluate on its default validation split. Following the standard protocol, we report top-1 accuracy at IoU thresholds 0.25 and 0.50 for ScanRefer (Acc@0.25 and Acc@0.50). For Multi3DRefer, we use the corresponding F1 scores(F1@0.25 and F1@0.50).

**Implementation Details.** We adopt the 3D visual encoder released with 3D-LLaVA (Deng et al., 2025) and build our model on LLaVA-1.5-7B (Liu et al., 2024a). The LLM used in TR$^2$ Semantic Structurizer is deepseek-v3-0324-64k. Instruction tuning is conducted on 8 NVIDIA RTX H100 GPUs with DeepSpeed's ZeRO Stage 1 optimization. Following common practice, we insert LoRA adapters (Hu et al., 2022) into the language model while freezing the remaining parameters of both the LLM and the visual encoder. Each GPU processes a mini-batch of two samples, and gradients are accumulated over eight steps, yielding an effective batch size of 128. We optimize the network with AdamW and apply a cosine-annealing schedule, starting from an initial learning rate of 2e-4. For the instance segmentation, we used the weights released by 3D-LLaVA (Deng et al., 2025) which are pretrained following the OpenScene (Peng et al., 2023) paradigm.

### 4.2 COMPARISONS WITH STATE-OF-THE-ARTS

We compare 3D-ATRES with both specialist models and LLM-based models. The specialist models include ScanRefer (Chen et al., 2020), BUTD-DETR (Jain et al., 2022), M3DRef-CLIP (Zhang et al., 2023), 3D-VisTa (Zhu et al., 2023), EDA (Wu et al., 2023), D-LISA (Zhang et al., 2024), MCLN (Guo et al., 2025), G$^3$-LQ (Wang et al., 2024), GPS (Jia et al., 2024), 3D-VLP (Yang et al., 2024) and ConcreteNet (Unal et al., 2024). To ensure a fair comparison, every LLM-based method is implemented with a language model of comparable size. The LLM-based models include 3D-LLM (Hong et al., 2023), PQ3D (Zhu et al., 2024c), ReGround3D (Zhu et al., 2024a), ReGround3D (Zhu et al., 2024a), Chat-Scene (Huang et al., 2024), LIBA (Wang et al., 2025c),

Table 1: Quantitative comparison of specialist models and LLM-based 3D referring methods on the ScanRefer and Multi3DRefer benchmarks. For ScanRefer we report the **Acc@0.25** and **Acc@0.5**, whereas for Multi3DRefer we follow the official protocol and provide the **F1@0.25** and **F1@0.5**. The best result in each column is highlighted in bold.

| Method | Venue | Base LLM | ScanRefer | | Multi3DRefer | |
|---|---|---|---|---|---|---|
| | | | Acc@0.25 | Acc@0.5 | F1@0.25 | F1@0.5 |
| *Specialist Models* | | | | | | |
| ScanRefer (Chen et al., 2020) | ECCV'20 | – | 42.4 | 26.0 | – | – |
| BUTD-DETR (Jain et al., 2022) | ECCV'22 | – | 50.4 | 38.6 | – | – |
| M3DRef-CLIP (Zhang et al., 2023) | ICCV'23 | – | – | 44.7 | 42.8 | 38.4 |
| 3D-VisTa (Zhu et al., 2023) | ICCV'23 | – | 50.6 | 45.8 | – | – |
| EDA (Wu et al., 2023) | CVPR'23 | – | 54.6 | 42.2 | – | – |
| D-LISA (Zhang et al., 2024) | NeurIPS'24 | – | 57.0 | 46.2 | – | – |
| MCLN (Guo et al., 2025) | ECCV'24 | – | 57.1 | 45.5 | – | – |
| $G^3$-LQ (Wang et al., 2024) | CVPR'24 | – | 56.9 | 45.6 | – | – |
| GPS (Jia et al., 2024) | CVPR'24 | – | – | 48.1 | – | – |
| 3D-VLP (Yang et al., 2024) | AAAI'24 | – | 51.7 | 40.5 | – | – |
| ConcreteNet (Unal et al., 2024) | ECCV'24 | – | 56.1 | 49.5 | – | – |
| *LLM-based Models* | | | | | | |
| 3D-LLM (Hong et al., 2023) | NeurIPS'23 | BLIP2-flant5 | 30.3 | – | – | – |
| PQ3D (Zhu et al., 2024c) | ECCV'24 | Vicuna-7B | 57.0 | 51.2 | – | – |
| ReGround3D (Zhu et al., 2024a) | ECCV'24 | BLIP2-flant5 | 53.1 | 41.1 | – | – |
| Chat-Scene (Huang et al., 2024) | NeurIPS'24 | Vicuna-7B | 55.5 | 50.2 | 57.1 | 52.4 |
| LIBA (Wang et al., 2025c) | AAAI'25 | – | 59.6 | 49.0 | – | 50.2 |
| Inst3D-LMM (Yu et al., 2025) | CVPR'25 | Vicuna1.5-7B | 57.8 | 51.6 | 58.3 | 53.5 |
| Video-3D LLM (Zheng et al., 2025) | CVPR'25 | LLaVA-Video 7B | 58.1 | **51.7** | 58.0 | 52.7 |
| 3D-LLaVA (Zhu et al., 2024b) | CVPR'25 | LLaVA-1.5-7B | 62.8 | 45.3 | 68.7 | 49.2 |
| 3D-ATRES (Ours) | – | LLaVA-1.5-7B | **65.3** | 50.7 | **72.7** | **55.7** |

Inst3D-LMM (Yu et al., 2025), Video-3D LLM (Zheng et al., 2025) and 3D-LLaVA (Deng et al., 2025). The quantitative results are summarized in Table 1.

**ScanRefer.** 3D-ATRES attains an Acc@0.25 of 65.3%, establishing a new state of the art across both specialist and LLM-based systems. Compared with the strongest specialist model, ConcreteNet (56.1% Acc@0.25), our approach yields a gain of +9.2%; compared with the best previous LLM-based model, 3D-LLaVA (62.8% Acc@0.25), we improve by +2.5%. At the stricter IoU threshold, 3D-ATRES achieves 50.7% Acc@0.5, which outperforms recent specialist solutions (e.g., 49.5% Acc@0.5 for ConcreteNet) and surpasses most LLM-based baselines by a clear margin (+5.4% over 3D-LLaVA, +9.6% over ReGround3D). The slight gap to the best Acc@0.5 scores, 51.7% by Video-3D LLM, can be attributed to the off-the-shelf segmentor we use to generate instance proposals; nevertheless, the consistently superior Acc@0.25 indicates that 3D-ATRES understands free-form language queries and complex 3D scenes more robustly than prior work.

**Multi3DRefer.** Multi3DRefer is considerably more challenging than ScanRefer because a single query can refer to multiple target objects and usually depends on fine-grained inter-object spatial relations. 3D-ATRES sets a new state of the art with an F1@0.25 of 72.7%, outperforming the specialist pipeline M3DRef-CLIP (42.8%) by 29.9% and the strongest previous LLM-based model 3D-LLaVA (68.7%) by 4.0%. At the stricter IoU threshold, we obtain F1@0.5 of 55.7%, surpassing the current LLM leader Inst3D-LLM (+2.2%) and all other baselines, including Video-3D LLM (+3.0%) and Chat-Scene (+3.3%). These results demonstrate that 3D-ATRES scales effectively to complex multi-object referring scenarios and exhibits enhanced precision in localising multiple targets.

### 4.3 ABLATION STUDY

Table 2 presents the ablation results on ScanRefer and Multi3DRefer validation sets.

**TR$^2$ Semantic Structurizer.** Removing the component leads to a performance degradation on the ScanRefer dataset, with Acc@0.25 dropping by 1.1%, which underscores the module's effectiveness. Notably, this specific ablation was intentionally omitted for the Multi3DRefer. While the TR$^2$

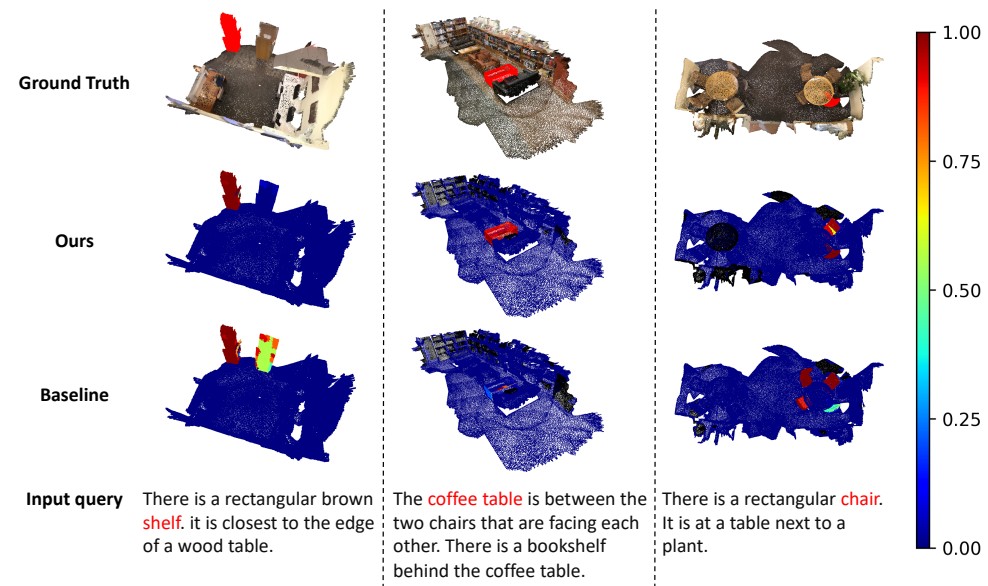

Figure 3: Visualization of 3D-ATRES's results in various ambiguities cases. The baseline model is 3D-LLaVA (Deng et al., 2025).

Table 2: Ablation study on the ScanRefer and Multi3DRefer validation sets.

| Method | ScanRefer | | Multi3DRefer | |
|---|---|---|---|---|
| | Acc@0.25 | Acc@0.5 | F1@0.25 | F1@0.5 |
| **Full 3D-ATRES (full modules)** | **65.3** | **50.7** | **72.7** | **55.7** |
| w/o TR$^2$ Semantic Structurizer | 64.2 | 49.6 | - | - |
| w/o Normal-Aware Spatial Alignment | 63.5 | 46.2 | 71.1 | 52.1 |
| w/o Annotation Ambiguity Penalty | 63.7 | 47.3 | 70.4 | 52.8 |
| Baseline (Deng et al., 2025) | 62.8 | 45.3 | 68.7 | 49.2 |

Semantic Structurizer demonstrates clear benefits on benchmarks like ScanRefer, we also recognize that its application is most effective when aligned with the underlying structure of the language data. The Multi3DRefer dataset presents a distinct challenge, as its expressions frequently encompass multiple objects. Applying a single-triplet parsing schema to these multi-target descriptions would represent an ill-posed formulation, potentially leading to information loss regarding secondary targets and consequently affecting recall. Therefore, to best leverage the strengths of our architecture in a data-aware manner, we adopt a principled approach: we utilize the structurizer to distill semantic relations from single-target expressions in ScanRefer, while for Multi3DRefer, we process the raw text directly to ensure all potential referents are considered. This tailored strategy allows our model to achieve optimal performance by adapting to the specific characteristics of each benchmark.

**Normal-Aware Spatial Alignment.** Removing the component hurts performance precisely where geometric consistency is measured most strictly. On ScanRefer, Acc@0.25 falls only from 65.3% to 63.5%, but Acc@0.5 drops much more sharply, from 50.7% to 46.2%. The same trend appears on Multi3DRefer: F1@0.25 decreases slightly from 72.7% to 71.1%, whereas F1@0.5 falls from 55.7% to 52.1%. The marked decline at the 0.5 IoU threshold confirms that explicit normal alignment is vital for geometric consistency. Without it, the model often fails to unify neighboring coplanar areas, like wall or tabletop segments, into single objects, producing fragmented masks that rarely meet the overlap requirements of the stricter metric.

**Annotation Ambiguity Penalty.** Removing the component lowers performance on both benchmarks and hits the cluttered Multi3DRefer set the hardest, with F1 dropping from 72.7% to 70.4%

and from 55.7% to 52.8%. ScanRefer also declines, from 65.3% to 63.7% (Acc@0.25) and 50.7% to 47.3% (Acc@0.5). Multi3DRefer's scenes often contain several similar objects described by imprecise language; assigning lower weights to these noisy labels prevents the model from overfitting to conflicting supervision. The smaller yet consistent decline on ScanRefer further demonstrates that explicitly modeling label uncertainty confers robustness even when annotation ambiguity is relatively mild.

The three components offer complementary benefits: TR$^2$SS enhances linguistic reasoning, NASA handles geometric variations, and AAP mitigates annotation noise. Together, they enable 3D-ATRES to effectively address the multi-source ambiguities in 3D RES.

### 4.4 ADDITIONAL EXPERIMENTS

**General LLM Capabilities.** Fine-tuning large language models for specialized tasks can lead to catastrophic forgetting, where general capabilities degrade. We show that 3D-ATRES not only avoids this but enhances overall 3D understanding. To assess the impact of our ambiguity-aware training, we benchmark 3D-ATRES against several state-of-the-art 3D multimodal models (LMMs) across various 3D vision-language tasks. As shown in Figure 4, our model (in red) outperforms or matches leading methods in benchmarks like ScanQA, Scan2Cap, and SQA3D, as well as ScanRefer and Multi3DRefer. The larger area covered by our model's plot reflects its more comprehensive capabilities. This indicates that our ambiguity-handling modules improve not only segmentation but also the integration of language and 3D geometry, resulting in a more capable and well-rounded 3D-LMM.

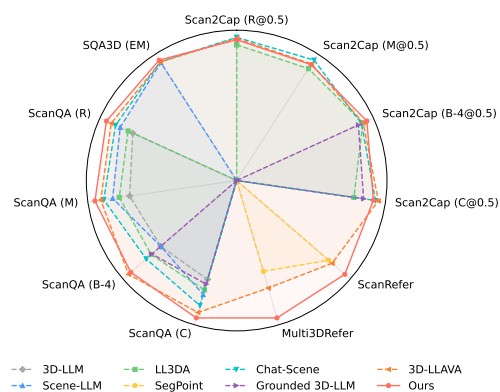

Figure 4: Comparison of our method with other state-of-the-art 3D LMMs. Our method outperforms competitors on most benchmarks.

**Visualization Result.** Figure 3 provides a qualitative comparison of the 3D-ATRES model ("Ours") against a baseline, demonstrating its superior handling of different types of ambiguity. In the left scenario, prompt ambiguity with distractors, the query asks for a "rectangular brown shelf" near a table. 3D-ATRES accurately segments the target shelf, while the baseline highlights both the shelf and a distractor. In the middle, the referring expression targets a "coffee table" with a complex spatial structure. 3D-ATRES accurately segments every component of the target object, whereas the baseline fails to ground the complete object. On the right, the query "rectangular chair next to a plant" is inherently ambiguous because several chairs in the scene satisfy these criteria, yet the ground-truth annotation labels only one of them. However, the ground truth annotates only one. The baseline wrongly segments additional chairs that do not fully meet the query constraints. In contrast, 3D-ATRES correctly isolates only the chairs that match the description, demonstrating the effectiveness of the Annotation Ambiguity Penalty (AAP) in learning from noisy supervision. More visualization can be found in Appendix A.2.

## 5 CONCLUSION

In this paper, we identified and addressed three fundamental ambiguity sources in 3D-RES: prompt, spatial, and annotation ambiguities. Our proposed 3D-ATRES integrates TR$^2$ Semantic Structurizer for structured language understanding, Normal-Aware Spatial Alignment for geometry-consistent feature learning, and Annotation Ambiguity Penalty for noise-robust training. Experiments on ScanRefer and Multi3DRefer demonstrate state-of-the-art performance, with significant improvements in handling ambiguous real-world scenarios. Our work highlights the importance of explicit ambiguity modeling in 3D vision-language tasks and opens avenues for extending these principles to broader 3D scene understanding problems.

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

# A  APPENDIX

---

**Algorithm 1:** Training Process with Annotation Ambiguity Penalty (AAP)

---

**Input:** Initialized network parameters $\theta$, fused features $\mathbf{F}$, labels $\mathbf{y}$, EM interval $T$, max_epoch, max_batch

**Output:** Optimized network parameters $\theta$

**for** $epoch = 1 : max\_epoch$ **do**

    **for** $batch = 1 : max\_batch$ **do**

        Forward prediction $\hat{\mathbf{y}} \leftarrow f_{seg}(\mathbf{F}; \theta)$;

        Calculate point-wise losses $\ell_i$ using Eq. (9);

        **if** $global\_iter \bmod T == 0$ **then**

            Update the GMM parameters $\Theta$ using Eq. (10);

        end

        Calculate the posterior probability $w_i$ using Eq. (11);

        Calculate $\mathcal{L}_{\text{AAP}}$ using Eq. (12);

        Calculate the total loss $\mathcal{L}_{\text{total}}$ using Eq. (13);

        Optimize the model parameters $\theta$ by $\mathcal{L}_{\text{total}}$;

        $global\_iter \leftarrow global\_iter + 1$;

    end

end

---

## A.1  GMM PARAMETER ESTIMATION FOR AAP

We introduced the *Annotation Ambiguity Penalty* (AAP) mechanism, where per-point binary cross-entropy (BCE) losses $\{\ell_i\}_{i=1}^N$ are modeled by a two-component Gaussian Mixture Model (GMM) (McLachlan & Peel, 2000):

$$p(\ell_i) = \pi_{\text{clean}} \cdot \mathcal{N}(\ell_i \mid \mu_{\text{clean}}, \sigma_{\text{clean}}^2) + \pi_{\text{noise}} \cdot \mathcal{N}(\ell_i \mid \mu_{\text{noise}}, \sigma_{\text{noise}}^2), \tag{14}$$

where $\pi_{\text{clean}}, \pi_{\text{noise}}$ are the mixture weights, $\mu_{\text{clean}}, \sigma_{\text{clean}}^2$ are the mean and variance of the clean-label loss distribution, and $\mu_{\text{noise}}, \sigma_{\text{noise}}^2$ are the mean and variance of the noisy-label distribution. The parameter set is denoted by:

$$\Theta = \{\mu_{\text{clean}}, \sigma_{\text{clean}}^2, \mu_{\text{noise}}, \sigma_{\text{noise}}^2, \pi_{\text{clean}}, \pi_{\text{noise}}\}.$$

We estimate $\Theta$ using the **Expectation–Maximization (EM)** (Dempster et al., 1977) algorithm every $T$ iterations during training to adapt to the evolving loss distribution.

### A.1.1  INITIALIZATION

At the first EM update:

  (i) We compute all $\ell_i$ in the current batch.

 (ii) We initialize $\mu_{\text{clean}}$ and $\mu_{\text{noise}}$ using K-means clustering with $K = 2$ applied to $\{\ell_i\}$.

(iii) The variances $\sigma_{\text{clean}}^2$ and $\sigma_{\text{noise}}^2$ are initialized as the sample variances within each cluster.

(iv) The mixture coefficients $\pi_{\text{clean}}$ and $\pi_{\text{noise}}$ are initialized from the relative cluster proportions.

### A.1.2  EM ITERATIONS

Given current parameters $\Theta^{(t)}$ at EM step $t$:

**E-step:**  For each point loss $\ell_i$, compute the posterior probability (responsibility) of belonging to the *clean* component:

$$\gamma_i^{\text{clean}} = \frac{\pi_{\text{clean}}^{(t)} \cdot \mathcal{N}(\ell_i \mid \mu_{\text{clean}}^{(t)}, \sigma_{\text{clean}}^{2\,(t)})}{p(\ell_i; \Theta^{(t)})}, \tag{15}$$

and similarly $\gamma_i^{\text{noise}} = 1 - \gamma_i^{\text{clean}}$.

**M-step:** Update the parameters:

$$\pi_{\text{clean}}^{(t+1)} = \frac{1}{N} \sum_{i=1}^{N} \gamma_i^{\text{clean}}, \quad \pi_{\text{noise}}^{(t+1)} = 1 - \pi_{\text{clean}}^{(t+1)}, \tag{16}$$

$$\mu_{\text{clean}}^{(t+1)} = \frac{\sum_{i=1}^{N} \gamma_i^{\text{clean}} \ell_i}{\sum_{i=1}^{N} \gamma_i^{\text{clean}}}, \quad \mu_{\text{noise}}^{(t+1)} = \frac{\sum_{i=1}^{N} \gamma_i^{\text{noise}} \ell_i}{\sum_{i=1}^{N} \gamma_i^{\text{noise}}}, \tag{17}$$

$$\sigma_{\text{clean}}^{2\,(t+1)} = \frac{\sum_{i=1}^{N} \gamma_i^{\text{clean}} (\ell_i - \mu_{\text{clean}}^{(t+1)})^2}{\sum_{i=1}^{N} \gamma_i^{\text{clean}}}, \tag{18}$$

$$\sigma_{\text{noise}}^{2\,(t+1)} = \frac{\sum_{i=1}^{N} \gamma_i^{\text{noise}} (\ell_i - \mu_{\text{noise}}^{(t+1)})^2}{\sum_{i=1}^{N} \gamma_i^{\text{noise}}}. \tag{19}$$

### A.1.3 INTEGRATION WITH AAP

After each EM update, the clean-label posterior $\gamma_i^{\text{clean}}$ serves as the confidence weight $w_i$ in the AAP loss:

$$\mathcal{L}_{\text{AAP}} = \frac{1}{N} \sum_{i=1}^{N} w_i \cdot \ell_i, \quad w_i \equiv \gamma_i^{\text{clean}}. \tag{20}$$

This weighting dynamically suppresses the contribution of samples likely to be mislabeled, without discarding any data.

### A.1.4 UPDATE FREQUENCY

We perform EM updates every $T$ iterations (e.g., $T = 200$ in our experiments), which provides a balance between the stability of the estimates and responsiveness to the evolving loss landscape.

### A.2 MORE VISUALIZATION

In Figure 5, we qualitatively analyse 3D-ATRES under three typical sources of ambiguity: referring, geometry and annotation.

**Prompt Ambiguity.** In Figure 5(a), the query jointly specifies a target object ("black table"), several referent objects ("gray armchairs" and "window"), and their spatial relations ("between" and "behind"). Equipped with the LLM-based Referring-Triple Parser, 3D-ATRES accurately resolves this composite expression and segments the unique ground-truth target. By contrast, the baseline model misinterprets the query and segments both the target and the distractors.

**Spatial Ambiguity.** In Figure 5(b), the off-the-shelf segmentor fragments the target shelf into several disconnected planar patches. Because the baseline model cannot associate these patches as belonging to the same object, it returns an incomplete mask. Leveraging the Normal-Aware Spatial Alignment module, 3D-ATRES explicitly aligns surface normals, merges the fragmented faces into a single instance, and ultimately delivers a coherent segmentation despite intra-object appearance variations.

**Annotation Ambiguity.** As shown in Figure 5(c), the query is satisfied by two distinct clothes dryers, yet the dataset provides a ground-truth mask for only one. Driven by the proposed GMM-based Label-Uncertainty Module, 3D-ATRES assigns high confidence to both legitimate candidates, exhibiting robustness to incomplete annotations and a deeper scene understanding instead of overfitting to potentially erroneous labels.

### A.3 INTERACTION TOOLS

In order to facilitate the display of the effect, we have created a visual interactive website that can visually showcase the usage process of our model.

(i) Prepare an indoor point cloud in Figure 6, you first need to upload a point cloud file, and the point cloud format should be ply. Alternatively, we can provide good point cloud examples

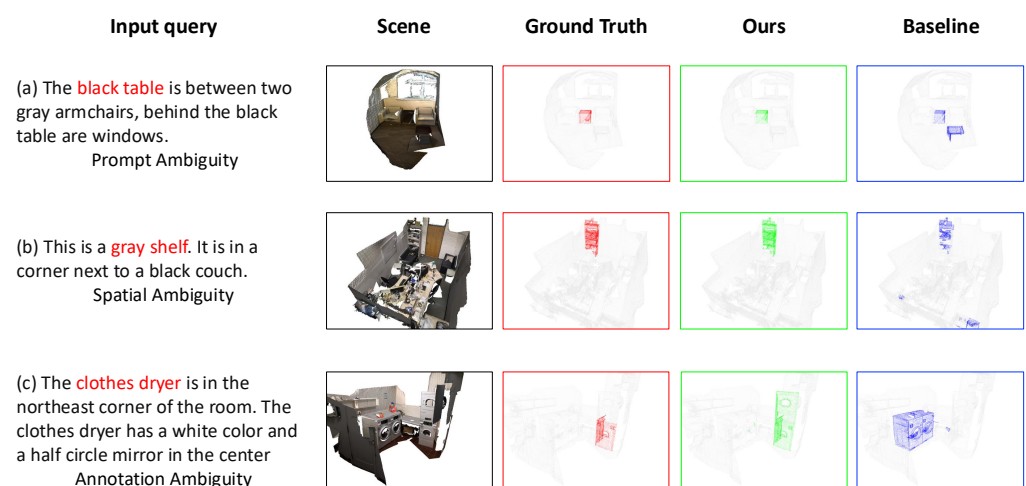

|  | Input query | Scene | Ground Truth | Ours | Baseline |

(a) The black table is between two gray armchairs, behind the black table are windows.
Prompt Ambiguity

(b) This is a gray shelf. It is in a corner next to a black couch.
Spatial Ambiguity

(c) The clothes dryer is in the northeast corner of the room. The clothes dryer has a white color and a half circle mirror in the center
Annotation Ambiguity

Figure 5: Visualization of 3D-ATRES's results in various ambiguities cases. The baseline model is 3D-LLaVA (Deng et al., 2025).

that we need to provide. After importing the point cloud model, it will appear in the left box, and users can freely drag and observe it.

(ii) Enter a description of finding an object in Figure 7, entering the description of the object to be searched for in the input field on the right, such as "The lamp between the two beds" in the example. In Figure 8, the model will display "Thinking", indicating that it is thinking. In Figure 9, after completing the thinking, the target point cloud found will be displayed in the box on the right.

(iii) Free dialogue Q&A in Figure 10, switching to chat mode, you can freely input scene questions, such as in this example asking how many beds are in the room, and the model will output results based on the scene.

## A.4 THE USE OF LARGE LANGUAGE MODELS (LLMS)

We disclose using LLMs solely to aid and polish writing—enhancing academic expression accuracy, argument coherence, and text logic. All core ideas, research designs, experiments, and conclusions are independently developed by the authors.

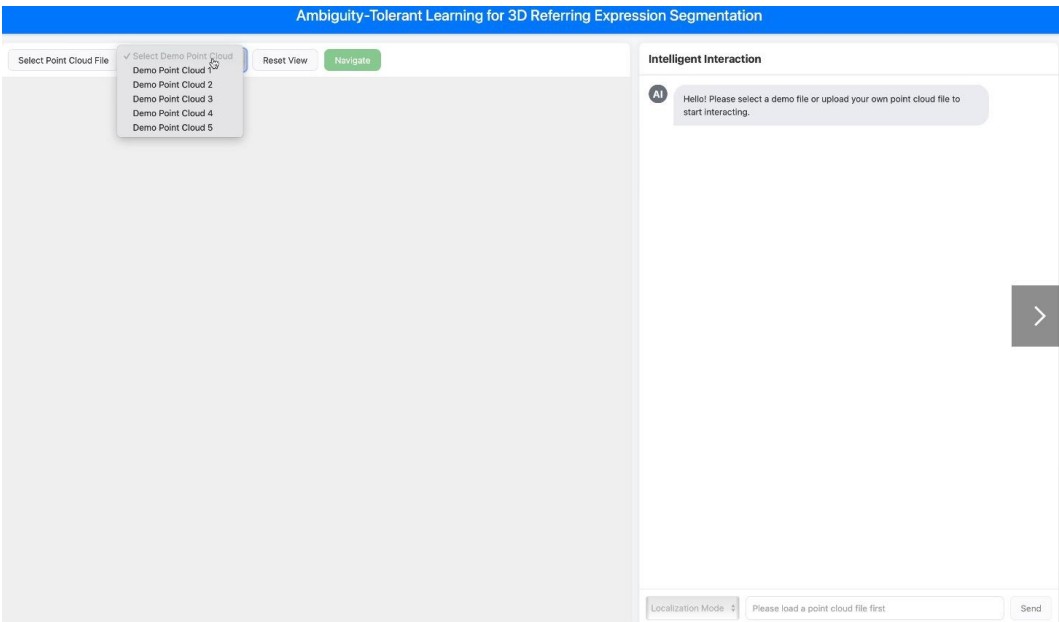

Figure 6: Point Cloud Upload Interface

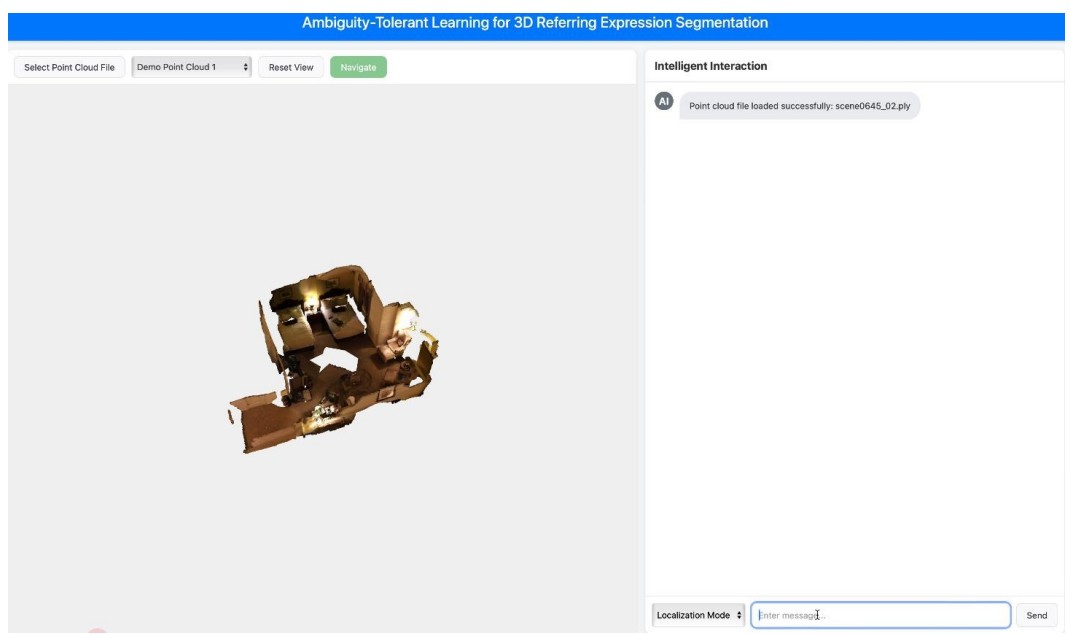

Figure 7: Target Object Description Input

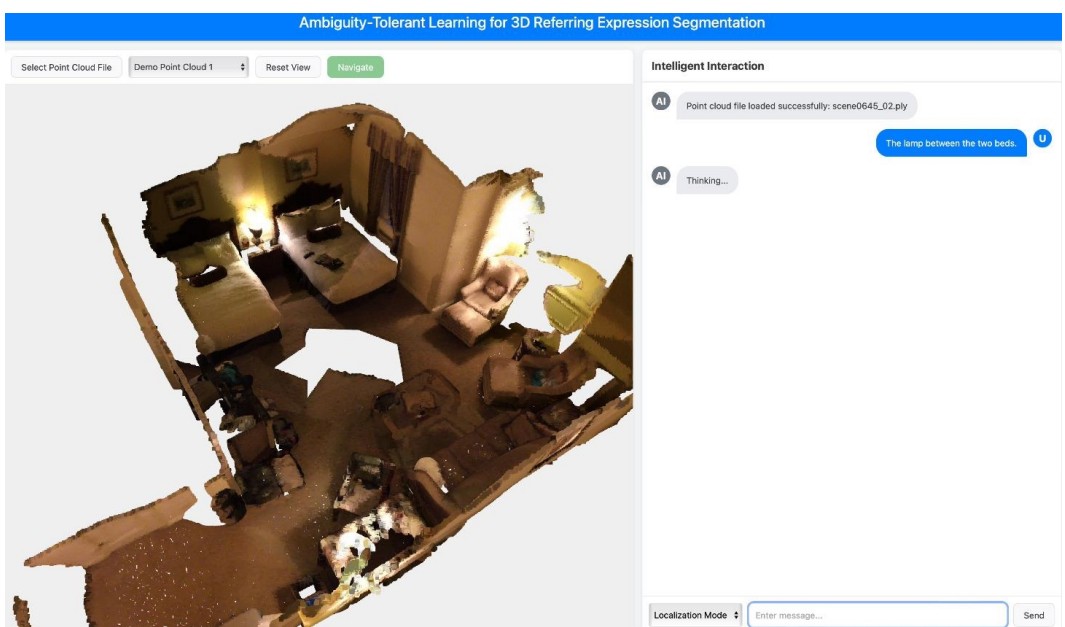

Figure 8: Model Processing Status

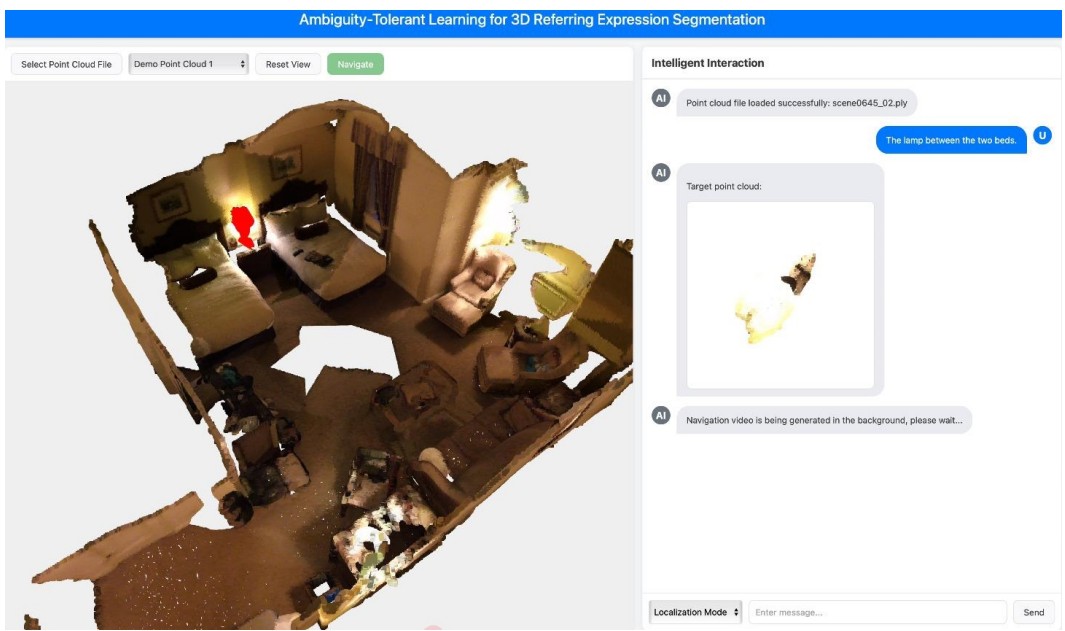

Figure 9: Search Result Display

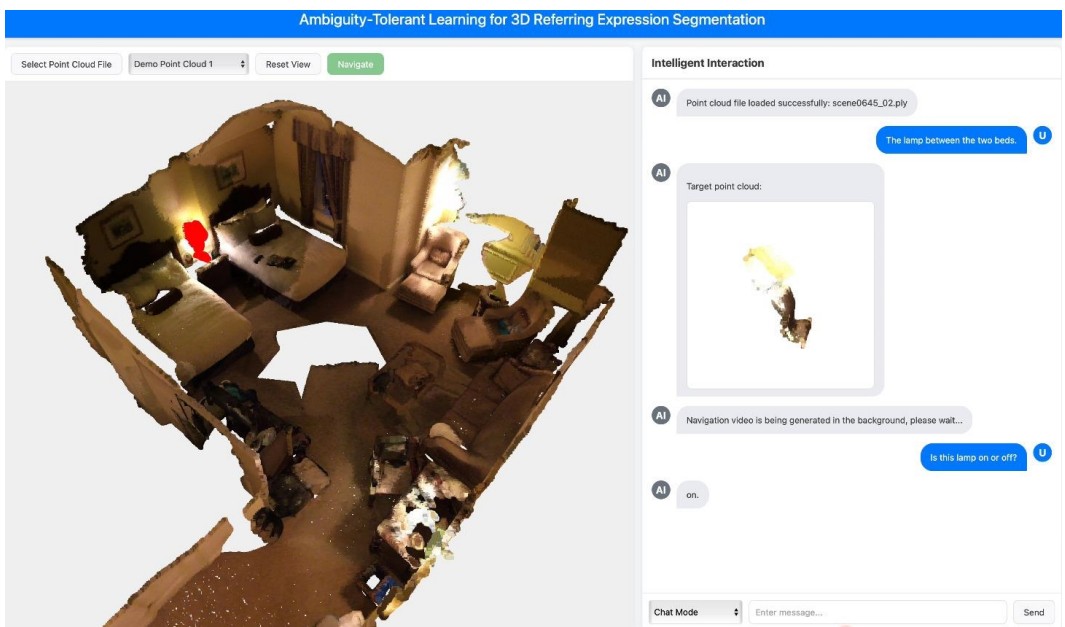

Figure 10: Free Q&A Interface

