# OpenReview forum: "3D-ATRES: Ambiguity-Tolerant Learning for 3D Referring Expression Segmentation"
_ICLR.cc/2026/Conference — Submitted to ICLR 2026_

### Official Review · Reviewer_9gzH · 2025-10-22

**Soundness:** 3
**Presentation:** 2
**Contribution:** 2
**Rating:** 4
**Confidence:** 4

**Summary:**

The paper proposes 3D-ATRES, a 3D referring expression segmentation (3D-RES) framework designed to be ambiguity-tolerant along three axes: prompt (linguistic), spatial (geometric), and annotation (label noise). To handle annotation ambiguity, the paper introduces a Gaussian-mixture-based weighting of per-point losses with EM updates.  On ScanRefer and Multi3DRefer the method shows gains over specialist and MLLM-based baselines, and ablations suggest all three modules help.

**Strengths:**

The paper clearly argues why ambiguity shows up in real scenes and ties each ambiguity to a concrete module. The triple representation is a reasonable bridge between language and 3D objects. The AAP component looks grounded in loss bimodality and standard EM for GMMs, implemented at a practical cadence. And the experiments cover two widely used datasets with standard metrics and show consistent gains; Table 1 and Table 2 are informative.

**Weaknesses:**

I think the omission of a TR2 ablation on Multi3DRefer leaves a gap. The text explains why the single-triplet schema is ill-posed for multi-target expressions, so TR2 is skipped on that dataset; see the paragraph under Table 2. That makes sense, but it also means we do not know if a multi-triple (or set-valued) variant could help. A simple fix is to try TR2 with a multi-relation expansion on Multi3DRefer and report F1 deltas. Even a negative result would bound the design choice.

The evaluation control across baselines could be tighter. Table 1 mixes numbers from prior papers and local re-runs, but the paper does not state whether all methods got the same input resolution, instance proposals, and instruction budget. This is especially relevant because you build on LLaVA-1.5-7B with an off-the-shelf instance segmentor and different choices move Acc@0.50 substantially; see the training/eval details in Sec. 4.1 and the discussion in Sec. 4.2 about the proposal generator affecting 0.5-IoU.  A practical patch is to include a small controlled re-evaluation under your harness (same proposals, same max turns, same resolution) for 2–3 strongest baselines.

The NASA loss would benefit from a sensitivity study. Right now $\phi(\theta)=\exp(-\theta)$ is fixed, with temperature $\tau=0.07$.  It is unclear whether a cosine-based or piecewise profile would work better on planar vs. curved parts. A simple sweep over $\phi$ forms and $\tau$ with Acc@0.25/0.50 would make the claim stronger.

The AAP mechanism needs more diagnostics beyond the main gains. The paper defines $p(\ell_i)$ as a 2-component GMM, computes $w_i=p(\text{clean}\mid \ell_i)$, and weights BCE.  Appendix A.1 reports EM details and an update interval $T$ (see A.1/A.1.4).  However, we do not see (i) how often the assignments flip during training, (ii) calibration of $w_i$ vs. actual label correctness on a small hand-audited subset, or (iii) the relative contribution of AAP in scenes with low ambiguity. Adding these would clarify when AAP helps and when it is neutral.

The generalization narrative in Fig. 4 is interesting but thin on protocol. The radar comparison suggests improved 3D-LMM capability across ScanQA/Scan2Cap/SQA3D, but the text does not specify whether backbones, proposals, and compute were matched.  It would help to include key settings and a short table with absolute numbers for at least one non-RES task.

There are a few clarity nits in the pipeline descriptions. In Fig. 2’s text and the surrounding paragraphs, the flow from superpoint pooling to LLaVA fusion to the segmentation head is compact; an explicit token/feature shape table (token counts, channels) would improve reproducibility (see Sec. 3 overview and Fig. 2 caption region).

I also feel it might be insufficient to see enough analysis or experimental items. I expect to see more analyses and insights.

**Questions:**

Can TR2 be extended to a multi-triple, set-valued representation for Multi3DRefer? A small prototype (extract multiple triples and aggregate) would tell us whether the current “skip TR2” design is optimal.

How sensitive is NASA to the choice of $\phi(\theta)$ and temperature $\tau$? A short sweep for Eq. (6)–(7) would help us see if exp(–θ) is the right profile.

For AAP, can you report a calibration plot of $w_i$ vs. true correctness on a hand-checked subset, plus the switching rate of component posteriors over epochs?

In Table 1, which rows were re-run locally under your harness, and which are taken from papers? If possible, please add a controlled subset where all methods share the same proposals, resolution, and turn budget.

---

### Official Review · Reviewer_B6iD · 2025-10-30

**Soundness:** 2
**Presentation:** 3
**Contribution:** 2
**Rating:** 4
**Confidence:** 5

**Summary:**

This manuscript identify three types of ambiguity in the 3D referring expression segmentation task and propose three corresponding modules to address these issues, ultimately achieving state-of-the-art performance on both the ScanRefer and Multi3DRef datasets.

**Strengths:**

- The paper is well-written and easy to follow.
- The proposed method achieves state-of-the-art performance on both the ScanRefer and Multi3DRef benchmarks.

**Weaknesses:**

1. Structured semantic representations are not suitable for all scenarios, as show in Section 4.3, it may harm the generalization of the LLM. This appears to be primarily a prompt engineering technique rather than a fundamental methodological contribution.
2. The motivation for employing an LLM for referring expression segmentation remains unclear. The method appears to utilize only the features from the [SEG] token for decoding, which could potentially be achieved through simpler alternatives such as multi-layer cross-attention mechanisms. The paper does not demonstrate or leverage the text generation capabilities that are characteristic of LLMs.
3. The overall contribution of the paper appears limited, with relatively straightforward proposed components (Prompt ambiguity has mentioned in early works like [1]. TR2 Semantic Structurizer resembles a prompt engineering trick, and $L_{NASA}$ is essentially a contrastive loss).
4. The ablation study is insufficient. To properly evaluate the contribution of normal vectors, the authors should ablate only the normal vector similarity score rather than removing the entire Normal Aware Spatial Alignment module.
5. The method for obtaining the normal vectors is not explained.

**Questions:**

1. Have the authors quantified the distribution of the three types of ambiguity in the dataset? This statistical analysis is crucial for assessing the significance and practical impact of this work.
2. Multiple approaches exist for decomposing text into structured triplets, such as Dependency tree [2, 3]. What is the rationale for using an LLM (DeepseekV3) for this decomposition? How does its effectiveness and efficiency compare with alternative methods?
3. In the Implementation Details, the authors claim to initialize their model using the 3D-LLaVA visual encoder and LLaVA-1.5-7B model. However, Section 4.4 presents performance results on multiple datasets on which the this paper was not trained. The authors need to clarify whether their method is trained from 3D-LLaVA's visual encoder and LLaVA-1.5-7B model or fine-tuned based on the pre-trained 3D-LLaVA model.
4. According to the description in Section 3.3, Normal-Aware Spatial Alignment is applied after the 3D U-Net, which appears to be independent of the [SEG] token. How are the gradients from this loss backpropagated to the LLM? Furthermore, given that the 3D U-Net is frozen during training, what role does this loss function actually play in optimizing the model?

# Reference:
[1] Wu Y, Cheng X, Zhang R, et al. EDA: Explicit text-decoupling and dense alignment for 3D visual grounding[C]//Proceedings of the IEEE/CVF Conference on Computer Vision and Pattern Recognition. 2023: 19231-19242.

[2] Schuster S, Krishna R, Chang A, et al. Generating semantically precise scene graphs from textual descriptions for improved image retrieval[C]//Proceedings of the fourth workshop on vision and language. 2015: 70-80.

[3] Wu H, Mao J, Zhang Y, et al. Unified visual-semantic embeddings: Bridging vision and language with structured meaning representations[C]//Proceedings of the IEEE/CVF Conference on Computer Vision and Pattern Recognition. 2019: 6609-6618.

---

### Official Review · Reviewer_fcLe · 2025-10-31

**Soundness:** 2
**Presentation:** 2
**Contribution:** 2
**Rating:** 4
**Confidence:** 5

**Summary:**

This paper introduces 3D-ATRES, a novel framework for 3D Referring Expression Segmentation (3D-RES) that explicitly models and mitigates three types of ambiguity, prompt, spatial, and annotation, which are often encountered in real-world 3D vision-language tasks. To address these challenges, the paper proposes: TR2 Semantic Structurizer: Converts free-form language into structured Target-Relation-Referent triples to reduce prompt ambiguity; Normal-Aware Spatial Alignment (NASA): Uses surface normals for geometric consistency to handle spatial ambiguity; Annotation Ambiguity Penalty (AAP): Learns from noisy/incomplete labels via a GMM-based reweighting scheme.
Experiments on ScanRefer and Multi3DRefer show improved accuracy over SOTA methods. The method is integrated with LLaVA-1.5-7B and evaluated with extensive quantitative and qualitative comparisons.

**Strengths:**

- Timely and Important Problem: The paper highlights an under-explored but practical problem, multi-source ambiguity in 3D-RES, and categorizes it into clear types (prompt, spatial, annotation).
- Empirical Gains: The model achieves strong results across multiple metrics and datasets (especially ScanRefer and Multi3DRefer), outperforming prior methods like 3D-LLaVA and ConcreteNet.

**Weaknesses:**

- Incremental Over 3D-LLaVA: Despite meaningful improvements, the overall architecture still heavily builds on 3D-LLaVA and OpenScene pipelines. The novelty is more in addressing annotation/training issues than architectural changes.
- Ambiguity Modeling via Heuristics: TR2SS relies on few-shot prompting using LLMs (e.g., DeepSeek-v3) for triple parsing, which is brittle and hard to standardize. No clear error analysis is given for failed or incorrect parses.
- Limited Evaluation Scope: Most results are confined to ScanRefer/Multi3DRefer. No experiments on more diverse or challenging datasets (e.g., Nr3D, SR3D, or ARKitScenes).
- Missing Details: It’s unclear how well TR2SS generalizes to non-English expressions or noisy text inputs. The annotation ambiguity experiments feel post-hoc—there is no ground-truth benchmark for label noise, so improvements may partly stem from overfitting cleaner regions.
- Lack of Broader Theoretical Insights: The components are intuitive but lack theoretical formulation or generalizability guarantees (e.g., what defines “ambiguity” in a measurable way beyond heuristic surface normals or label losses?).

**Questions:**

See weaknesses

---

### Official Review · Reviewer_bs6c · 2025-11-01

**Soundness:** 3
**Presentation:** 2
**Contribution:** 2
**Rating:** 4
**Confidence:** 4

**Summary:**

This paper introduces **3D-ATRES**, a framework for 3D-RES that identifies and tackles three key ambiguities: 1) prompt ambiguity (vague language), 2) spatial ambiguity (fragmented 3D scans), and 3) annotation ambiguity (noisy labels). To address these, the authors introduce three modules:
1. TR$^2$ Semantic Structurizer (TR$^2$SS): A pre-processing step that uses an LLM to parse queries into structured (Target, Relation, Referent) triples.
2. Normal-Aware Spatial Alignment (NASA): A novel geometry-aware contrastive loss that uses surface normal deviations to correctly group disconnected object parts.
3. Annotation Ambiguity Penalty (AAP): A GMM-based loss re-weighting scheme to identify and down-weight points with noisy labels.
Experimental results on ScanRefer, Multi3DRefer, and a small subset of Nr3D show consistent improvements over recent baselines such as X-RefSeg3D, 3DGraphLLM, and 3D-LLaVA. In addition, the ablation studies indicate that each ambiguity-specific module contributes incremental gains.

**Strengths:**

1. Explicitly exploring three distinct ambiguity sources provides a coherent perspective on why 3D-RES methods fail and how to correct them.
2. Each component of the proposed method has a clear target (language, geometry, or annotation ambiguity) and can, in principle, be integrated into other architectures.
3. Extensive experimental results and ablation studies are provided.

**Weaknesses:**

1. The technical novelty of 3D-ATRES appears to be incremental. The overall method is an integration of known techniques. TR$^2$SS resembles scene-graph or relation-aware parsing already explored in ViL3DRel, X-RefSeg3D, and 3DGraphLLM. NASA extends standard geometry-aware contrastive alignment found in earlier 3D vision works (e.g., ViL3DRel orientation encoding, GPA-3D). AAP parallels prior noise-robust point-cloud segmentation via GMM reweighting. For example, see “Robust Point Cloud Segmentation with Noisy Annotations” by  Ye, X. et al. (TPAMI 2023).
2. The complexity of the final system could be an issue. The method is not a single elegant model but rather a 3D-MLLM (3D-LLaVA), which is already a large system, with another massive LLM (deepseek-v3) used as a pre-processor (TR$^2$SS), plus a complex new geometry loss ($L_{NASA}$), plus a GMM-based noisy label module ($L_{AAP}$). This is a very heavy-duty, multi-stage pipeline, which may be difficult to train, reproduce, and deploy.
3. The “normal-aware” formulation is only normal-weighted rather than jointly embedding normal information with feature representations. The paper should clarify why this relatively simple weighting scheme is preferable or more effective than a joint normal–feature embedding, and under what conditions such a design choice offers tangible advantages.
4. In the experimental results, the reported performance gains over 3D-LLaVA and 3DGraphLLM are within 1–3 percentage points. However, since no standard deviations or multi-run averages are provided, it remains uncertain whether these improvements are statistically significant. Consequently, the claimed “ambiguity-tolerant” advantage may be somewhat overstated in the absence of supporting statistical evidence.
5. The ablation for TR$^2$SS is omitted on Multi3DRefer. Although the stated justification that its formulation is ill posed for multi-target queries is reasonable, it would be informative to know what actually occurs if the module is applied in such cases. Does the performance degrade sharply, or does it simply fail to provide additional benefit? More importantly, this limitation highlights a key weakness of the framework because one of its three core modules is not general-purpose and functions only in single-target scenarios.
6. It would be insightful to report failure cases, e.g., where TR$^2$SS mis-parses a query or where normals are noisy.

**Questions:**

The authors are suggested to respond to those raised in **Weaknesses.**

---

### Meta-Review · Area_Chair_UEuZ · 2025-12-24

**Summary:**

The paper initially received negative reviews, with all scores being a 4.

Reviewers identified several weaknesses in the paper, including limited novelty and contributions, insufficient supporting statistical evidence for the claimed advantage, and inadequate ablation studies. The authors did not provide a rebuttal to these concerns.

The area chair agrees with the reviewers' evaluation and recommends rejecting the paper.

**Reviewer Concerns:**

The authors did not respond to the reviewers' concerns.

**Reviewer Scores:**

The area chair expects the reviewers to maintain their initial scores at 4.

---

### Decision · Program_Chairs · 2026-01-26

Reject